# Smartphone Applications Designed to Improve Older People’s Chronic Pain Management: An Integrated Systematic Review

**DOI:** 10.3390/geriatrics6020040

**Published:** 2021-04-08

**Authors:** Margaret Dunham, Antonio Bonacaro, Patricia Schofield, Liz Bacon, Fotios Spyridonis, Hadi Mehrpouya

**Affiliations:** 1College of Health Wellbeing and Life Sciences, Sheffield Hallam University, Sheffield S1 1WB, UK; patricia.schofield@shu.ac.uk; 2School of Nursing, Anglia Ruskin University, Chelmsford CM1 1SQ, UK; antonio.bonacaro@aru.ac.uk; 3Abertay University, Dundee DD1 1HG, UK; l.bacon@abertay.ac.uk; 4School of Computing and Mathematical Sciences, University of Greenwich, London SE10 9LS, UK; f.spyridonis@greenwich.ac.uk; 5School of Design and Informatics, Abertay University, Dundee DD1 1HG, UK; h.mehrpouya@abertay.ac.uk

**Keywords:** telehealth, chronic pain, mHealth, smartphone apps, chatbot

## Abstract

(1) Background: Older people’s chronic pain is often not well managed because of fears of side-effects and under-reporting. Telehealth interventions, in the form of smartphone applications, are attracting much interest in the management of chronic diseases, with new and evolving approaches in response to current population demographics. However, the extent to which telehealth interventions may be used to promote and effect the self-management of chronic pain is not established. (2) Aim: To provide an objective review of the existing quantitative and qualitative evidence pertaining to the benefits of smartphone applications for the management of chronic pain in older people. (3) Methods: A literature search was undertaken using PubMed, Medline, CINAHL, Embase, PsychINFO, the Cochrane database, Science Direct and references of retrieved articles. The data were independently extracted by two reviewers from the original reports. (4) Results: This integrative systematic review identified 10 articles considering smartphone applications related to self-management of chronic pain among older adults. (5) Conclusions: It is important for future research to not only examine the effects of smartphone initiatives, but also to compare their safety, acceptability, efficacy and cost–benefit ratio in relation to existing treatment modalities.

## 1. Introduction

The increasingly ageing population, with associated chronic health comorbidities, poses a major global public health challenge [1]. Chronic pain is a long term condition that is a significant comorbidity of ageing associated with long-term conditions, and it affects at least 50% of people aged 60 years and older [2,3]. Older people make up an increasing proportion of all healthcare consumers and the incidence of complex health problems for the “old-old” is growing correspondingly. Thus, older people are more likely to access healthcare and the provision of their care is potentially costly to society.

The current COVID-19 pandemic has severely affected normal healthcare delivery and highlighted a pressing need to consider alternative approaches to supporting older people with their healthcare needs [4]. Physical access to a service should not prevent access to healthcare services. Older people may experience difficulties travelling to a hospital or GP and the possibility of telehealth initiatives may offer an opportunity for accessing support without having to leave home. With increasing pressure on healthcare provision and community infrastructure to deliver care and services, digital health technologies utilising existing mobile phone technology may offer alternative approaches to safely engaging with and supporting, older and vulnerable populations to facilitate self-care. Therefore, it is paramount to consider pursuing such alternative strategies to empower both older people with chronic pain and healthcare professionals, and to encourage the adoption of advanced technologies.

## 2. Background

Chronic pain is a significant cause of disability; estimates of its effects are largely based on studies of adult working populations [5]. The prevalence of chronic pain in the UK and Europe may vary from 35.0% to 51.3% of the adult population [5,6,7]. The socioeconomic implications of chronic pain are estimated to be considerable [5,8,9]. As the prevalence of chronic pain increases with age, the management of chronic pain is acknowledged as a significant and growing problem for the ageing population [10,11]. There are also significant challenges to resourcing a healthcare workforce to support the global aging population [12,13].

Furthermore, the effect of the COVID-19 pandemic has rightly focused healthcare provision on critical care delivery for respiratory distress, along with the safety of healthcare workers through prevention of infection. This current pandemic, along with the increasingly ageing population, has highlighted a pressing need to consider alternative innovative ways of working which minimise direct patient contact [14] or the need for older adults to travel distances to their care provider. The requirement for social distancing has temporarily affected many aspects of healthcare provision, including elective surgery and support for the management of chronic disease. Additionally, pain services in the UK are currently severely disrupted by the current pandemic [4].

Digital technology in the form of mobile health solutions has the potential to support wellbeing, and address some of the health and social care needs of an ageing population, particularly those living with long-term conditions with increasing evidence of adoption and acceptance [15,16,17,18,19,20,21]. The popular myth is that older people are resistant to using new technology [22]. However, chronological age is not the sole determinant of technological acceptance and adoption, as education and socioeconomic factors are possible influences [15,16,17,18,19,20,21,22,23]. Ageist stereotypes, healthcare professionals’ scepticism and negative assumptions may also form considerable barriers for equal access to healthcare technology [24,25,26,27].

Innovations in the form of smartphone applications have the potential to address some of the healthcare needs as highlighted in the pandemic. In the UK, 70% of adults over 55 own a smartphone which makes this platform an appropriate channel to support the health needs of older adults [28]. The first smart phones with operating systems capable of supporting healthcare applications or “apps” appeared in 2008 [29]. Since then, there is increasing evidence of smartphone interventions in the management of chronic diseases [30,31,32,33].

A review of smartphone apps for pain self-management available for download in 2012 found 220 apps, half of which focused on chronic pain [34]. A similar review of publicly available self-management apps for older adults with arthritic pain noted that identified pain apps had no evidence of formal research assessment and were not aligned with the evidence base for pain management [35]. Concerns have also been noted about the lack of regulatory oversight for patient welfare, data use and safety [36,37].

The sheer number of commercially available healthcare smartphone applications has been acknowledged but there is scant evidence for their effectiveness, and many are apparently developed without input from health professionals, evidence of clinical effectiveness, regulation or monitoring of use [34,38,39].

Given the high penetration rate of smartphones in the population, it is important to investigate the current adoption and use of suitable smartphone apps by older adults to inform future healthcare practice.

To our knowledge, this review is the first to consider the strength and quality of evidence for use of smartphone applications to support self-management for older people experiencing chronic pain.

## 3. Aims

To consider the existing evidence pertaining to the benefits of smartphone applications for the management of chronic pain in older people.

Objectives: To appraise the evidence for use of smartphone applications on self-care, to support chronic pain management for older people. To identify the elements of self-care support for chronic pain that can be delivered by via smartphone intervention.

## 4. Methods

### 4.1. Design

An integrative systematic review methodology was chosen to provide a comprehensive understanding of the healthcare problem and to support the inclusion of both qualitative and quantitative studies [40].

### 4.2. Search Strategy

An initial and updated literature search was conducted in May 2020 and December 2020 respectively, using a combination of key terms related to Pain, Older People and Telehealth initiatives in the databases PubMed, Medline, CINAHL, Embase, PsychINFO, Cochrane Database and Science Direct. An information scientist supported the development of the initial search terms and identification of MeSH terms. Two reviewers (MD and AB) performed initial searches independently.

### 4.3. Eligibility Criteria

Studies were selected according to the Population, Intervention, Comparator, Outcome(s) of interest, and Study design (PICOS) framework [41] as noted in Table 1.:

MeSH alternatives and associated terminology were applied and adapted for each database search to maximise inclusion of relevant data. The age parameters were adapted to include those studies reporting a mean age of 60 years to increase potential research papers. This is a selection of keywords used to search in PubMed:


*(“Pain” [MeSH Terms]) AND (“aged”[All Fields] OR “aged”[MeSH Terms] OR “elderly”[All Fields] OR “old”[All Fields] OR “seniors”[All Fields] OR “senior”[All Fields]) AND (“telemedicine”[MeSH Terms] OR “telemedicine”[All Fields] OR “tele-medicine”[All Fields] OR “telehealth”[All Fields] OR “tele-health”[All Fields] OR “mhealth”[All Fields] OR “m-health”[All Fields] OR “ehealth”[All Fields] OR “e-health”[All Fields])*


A hand-search of the reference lists of included studies, relevant reviews, national clinical practice guidelines and other relevant documents was undertaken. Only published English-language studies were included in this review.

The review identified a total of 506 papers after duplicates were removed. These included 362 from PubMed, Medline, CINHAL, Embase, the Cochrane Database and Science Direct combined, 142 PsychINFO and 2 from other sources using smartphone technology for various health conditions. Titles and abstracts of papers were screened by MD and AB to identify if they met the inclusion criteria. Following assessment of the abstracts, full text versions of 38 papers were retrieved for further scrutiny by the review team. Figure 1 shows the selection process.

No quantitative studies were identified that exclusively considered the use of smartphone app technology with older adults experiencing chronic pain.

### 4.4. Quality Assessment

Due to the heterogeneity in the designs of the identified studies, a recognised mixed methods critical appraisal tool was utilised to assess the methodological rigour of each study [42]. All papers were appraised to ascertain their methodological quality, the assessment criteria related to the abstract/title, introduction/aims, methods and data, sampling, data analysis, ethics and bias, findings/results, transferability/generalisability and implication/usefulness to determine their suitability for inclusion. No studies were excluded based on the level of methodological rigour.

### 4.5. Data Abstraction and Synthesis

The papers included quantitative and qualitative study designs, so an integrative methodology was applied to the retrieved studies. Extracted data included research question, design, sample size, setting and main findings (Table 2). There was great variability between studies and thus it was not possible to conduct any informative analysis of available quantitative data. Due to this heterogeneity, a narrative synthesis of the literature was conducted, to consider the findings of each study and take an iterative approach to the development of common themes [42]. A thematic analysis was subsequently conducted by the first author exploring the relationships and commonalities within and between studies.

## 5. Results

The review identified 10 papers that met the inclusion criteria. A summary of selected papers is included in Table 1. Within the included papers there is one pilot for an RCT, one quasi-experimental study, two mixed methods studies and six qualitative studies.

### 5.1. Study Characteristics

The quasi-experimental study considered the potential for a web-based pain management programme to improve clinical outcomes and reduce healthcare costs [51]. The use of a pain assessment programme via a smartphone was found to have potential clinical utility in a pilot mixed methods study [43]. Similarly, the potential of an mHealth smartphone intervention for supporting older remote rural communities had potential to supplement and complement existing chronic pain services [46]. A pilot of a “virtual pain coach” for older people with osteoarthritis considered its potential to enhance communication [49].

The identified qualitative studies were mainly exploratory, endorsing the need to include older service users in design and development to meet the needs of older people. The opinions of older people with chronic pain regarding an mHealth intervention via a smartphone suggested the importance of information sharing, including education to support self-care and self-administration of analgesia [53]. Five of the qualitative studies were pilot or feasibility studies with either service users or health professionals [44,45,47,48,50].

Due to the heterogeneity of study parameters and outcomes, a narrative synthesis and thematic analysis was conducted [42,53]. Four main themes were extracted from the selected papers, which were not mutually exclusive. The themes are: (1) The potential benefits and serviceability for older people; (2) Clinical and service user involvement in development; (3) Support or perceived barriers for clinicians and service users in their effective use; and (4) Use of data.

### 5.2. The Potential Benefits and Serviceabilityfor Older Adults

The identified qualitative studies mostly addressed the needs of the user, the older person, the service user or the healthcare professional informing the plan of care. The quantitative and mixed methods studies considered the potential economic and clinical impact of smartphone apps.

Chronic pain apps have the potential to be valuable tools for self-management, however not all older people will find this approach useful or relevant. A 2013 US randomised pilot study with 23 older adults (>60 years) tested the utility of a virtual pain coach administered via an app [49]. There were no discernible benefits for the intervention group, and while the virtual coach was well received and communication appeared to be enhanced, there were only 23 participants. Furthermore, there was no information regarding the prior technical abilities of any participants.

A north American qualitative study considered older people’s perceptions of and engagement with mHealth via a smartphone to support their chronic pain management [52]. The participants acknowledged the help with self-care, management of opioid medication, pain communications and potential for social interaction that a smartphone mHealth intervention could provide.

A 2014 study explored the potential barriers for health professionals in the application of telemedicine via smartphones in the provision of care for older adults with chronic pain [48]. The issues arising from the focus groups related to the requirements for support for both health professionals and service users in implementation of telemedicine interventions and the need to establish utility. Noted potential benefits included reduced travel for older people to access support and savings in clinical time.

A more recent Australian study of older people (*n* = 18), considered the potential benefits to current self-management practices and the need for personalisation to align with each service user’s needs [45]. They also noted the possibility of harm, as evidenced by some who expressed apprehension related to the potential for amplification of anxiety, negative emotions and catastrophisation of the pain experience.

An evaluation of a UK digital pain management programme was the only study to claim a significant potential cost benefit for service users utilising their novel version of a standard intervention [51].

### 5.3. Clinical and Service User Involvement in Development

Smartphone apps should be developed with a clear evidence base and relevant content with flexibility to tailor content to the service user. Engagement with the service user in the development and construction of smartphone or telehealth initiatives is a recurring issue. Equally, an understanding of their utility and potential should be shared by the health professionals and carers. Interviews with older adults living with chronic pain in rural Scotland, as part of a larger survey about the use of technology in healthcare, demonstrated the acceptability and potential for use of mHealth interventions to supplement care [46].

A team from Cornell University USA [48,50] considered the potential for use of telehealth, a smartphone initiative, in the management of chronic pain in older people, engaging initially with older people followed by primary care providers. The focus groups with older people (mean age 76 years) in New York, USA, described the potential for telemedicine for chronic pain, described as mHealth, utilising smartphones and tablets [50]. The possibility of usefulness was acknowledged by most participants however, some noted potential barriers including cost, the expense of any device, privacy of data and sensory limitations to utility. Subsequent engagement with care providers supported an interest in the usefulness of telemedicine [29].

A UK chronic pain team and pain researchers developed a digital online version of an existing and well established pain management programme (PMP) [54] and recruited 738 adult participants with chronic pain [51]. The intervention arm was supported by telephone and email communications and 179 service users completed the digital PMP. Although the age range of participants was 18 to 92 years, the older participants were most likely to complete the study, possibly giving some credibility to its utility in older populations.

Usability was noted as a major strength of a smartphone application in a 2019 study [43]. To ensure ease of use, the team held development workshops with members of Keele University’s Institute for Primary Care and Health Sciences (IPCHS), a Research User Group (RUG) and a supporting Clinical Advisory Group (CAG). Clinicians and service users were involved in the development of several iterations of the application, “developed by patients for patients” and therein lies its strength. The Keele Patient and Public Involvement and Engagement (PPI) group included older people who recorded pain more often than younger participants. The resulting app was liked by both patients and GPs, and early testing showed promising results in terms of face and content validity, acceptability and clinical usefulness, however numbers in the study were small (*n* = 21). Although the study was not targeted at older people, the median age for participants was 60 years (range 50–70 years).

### 5.4. Support or Perceived Barriers for Clinicians and Service Users in Their Effective Use

The perception of benefits or barriers to any smartphone mHealth initiative should be understood to ensure engagement. Older users found it beneficial to have associated input from their clinicians to establish and reinforce the use of technology or a pain self-management app [43,45,46]. Clinicians themselves expressed concern that not all service users or the clinicians responsible for their care would be willing to embrace new technology [43,44,48]. The importance of clinicians’ engagement with and confidence in the use of the app was presented as important for some of the Australian older people interviewed in Bhattarai’s study of older people using an app for chronic arthritic pain [25,45]. There was some general apprehension about the use of apps and concerns about the potential for pain to become an overly negative focus [45].

Levine noted the importance of design and data presentation; in particular, they noted potential “information overload” to be the most important barrier to device implementation [48]. Other practical concerns were the risk of litigation and the potential for additional cost if a more modern mobile phone might be required. Similarly, the Keele research group noted the importance of using plain English and having clear and accessible instructions [43]. Another practical consideration noted in one of the qualitative studies was the limited availability or inconsistency of mobile phone signal, broadband or internet services in some rural areas [46].

One study focused on identifying some of the practical considerations of use and barriers to the use of a smartphone or mHealth device [50]. The perceived barriers included device battery failure or other technical malfunctions, prohibitive cost, challenging technology and reduction in human interaction. [50]. However, these responses were not the responses of the majority of the older people willing to use an mHealth initiative.

Clinicians need to be well supported to review, identify and recommend pain self-management apps suitable for their older clients/patients. The potential for a pain assessment app for older people and its utility in pre-hospital assessment was tested with paramedic students in the UK [47]. Informed by the Abbey Pain Scale [55], Docking and team noted the usefulness of the app for assessing pain in people with dementia. However, this study was limited to focus groups away from the clinical setting and the team acknowledged the need to ensure involvement of service users in any further development.

Bhattarai and colleagues noted the clinician’s perspective on the use of a chatbot app [44]. The criticality of such a vehicle in facilitating the pain self-management process of older people was acknowledged, particularly because they form a significant and growing proportion of the population requiring access to pain services. Seventeen Australian primary care and allied health clinicians were interviewed about their views on the integration of apps into older people’s pain self-management strategy. The potential for empowerment and increased engagement was noted. However, some challenges and possible barriers were considered. These challenges included the clinicians’ time commitment in familiarising themselves with the app and introducing it to individual patients. Overall, this small study acknowledged the potential benefits if individualisation could be achieved within the software.

### 5.5. Use of Data

Data generated from app use could be utilised to improve monitoring and management of older people’s pain through enhanced systems, policies and procedures. In complementary studies, Australian researchers concluded that access to patient data, captured from pain apps, could enhance and improve the monitoring and management of older people’s healthcare needs [25,36,44,45]. Furthermore, clinicians could benefit from “health systems-level” policies and procedures as informed by the capture and use of such data. It is noteworthy that the participants in one of the qualitative studies were concerned about data privacy and in particular how their data might be used and safeguarded [50].

## 6. Discussion

Despite the potential breadth of the inclusion criteria, we found insufficient data from which to draw any strong conclusions regarding the evidence for smartphone use and telehealth for older people. The findings of this review reflect the broader absence of research to support older people living with chronic pain and other long-term conditions. There are no large-scale studies focusing on the particular needs of older people with chronic pain.

Our review suggests that there is an extensive level of development in mHealth and telemedicine use in management of chronic pain. Pain data from participants has the potential to be used to inform assessment and management of chronic pain by health professionals [43,49,50,51], improve individual understanding and self-management [44,45,47,48] and to provide external support. Although some of the works reviewed here did not focus exclusively on the needs of older adults, there is considerable potential for smartphone telehealth initiatives to support active healthy ageing populations and as an alternative mode of providing care to older adults living with long term conditions and chronic disease [14].

Telehealth/telemedicine interventions via smartphones have been gradually adopted in the UK healthcare system. In 2019 the National Institute for Health and Care Excellence (NICE) published standards for digital health technology [56]. Options for remote working, such as smartphone apps to support and manage care, should be considered as helpful ways to support regular care during a time of crisis and to benefit future care provision generally. Advantages of smartphone telehealth interventions include anonymity and reduced waiting time, and flexibility in terms of time and location of use [57]. The importance of data privacy and confidentiality should be acknowledged; to enable access by healthcare professionals the use of “closed systems”, encryption and password-protected access to any data should be considered. However, communications delivered via smartphone apps have the potential to support people with long-term health conditions in their everyday life, as they are cheap, informal and popular [58].

The World Health Organisation has recently launched a smartphone (telehealth) initiative to support the care and diverse needs of older people [59]. Evidence for the feasibility of telehealth has been noted in those living with long-term conditions including chronic lung disease, diabetes and heart failure [60]. There is growing evidence for the potential benefit of telehealth interventions as both resourceful and effective for pain related problems [61,62,63]. Older people can be very receptive to telemedicine as an augment to managing pain [46] and they are, in the main, technologically literate and receptive to the potential use of mHealth technologies in the industrialised world [64]. Associated benefits include support for medication management, enhanced communication with providers and reduction in feelings of isolation [63]. Tele monitoring via a smartphone app also has the potential to improve patient safety through rapid communication of unwanted events [65].

The NHS has approved some applications, which are targeted at meeting older people’s healthcare needs. The principle for a web-based “digital intervention” to manage chronic pain, the app “Pathway through Pain”, has been established with potential to improve healthcare outcomes and reduce the cost of care [51]. However, there is currently no approved app for the particular complex needs of older adults, and those living with multiple comorbidities, with chronic pain in the UK. A recent feasibility study has established the acceptability and potential benefit of a smartphone chatbot interface for the self-management and support for all kinds of chronic pain in older adults [66].

For people living with chronic pain, telemedicine in the form of a smartphone application or chatbot has many potential benefits as part of a new model of healthcare services, in terms of monitoring to improve practice, support for self-care, efficacy and resource utility.

### 6.1. Review Limitations

The apps identified in this review are a small proportion of commercially available smartphone applications. This is partially due to the large number of apps, and developments in telehealth that have been driven by the commercial potential for data mining. Another contributing factor to the limited scope of this review is that some of these apps are not currently available or were developed for research purposes, including feasibility studies.

The heterogeneity of papers identified reflects the variety of available apps developed for pain, particular painful conditions or populations. No information about personalisation options was noted.

### 6.2. Implications for Practice

Chronic pain smartphone apps have the potential to be valuable tools for self-management, however not all older people will find this approach useful or relevant. According to the United Nations (2019), the population of adults aged 65 years or over is growing faster than all other age groups [67]. The population of older adults is expected to double by 2050 to 1.5 billion individuals. As individuals age, they experience barriers and may have difficulties in vision, hearing, cognition and mobility that may affect access to healthcare technology [68,69]. In order to improve the accessibility issue, service users should be co-partners in the development and design of such technological approaches. Smartphone apps should be developed with a clear evidence base and relevant content with flexibility to tailor content to the service user. In addition, there is a need for more support for clinicians to review, identify and recommend pain self-management apps suitable for their older clients/patients. Older users may find it beneficial to have associated input from their clinicians to establish and use a pain self-management app. Finally, appropriate safeguards and governance are required for management of health data and its use across the health system to improve monitoring and management of older people’s pain through enhanced systems, policies and procedures.

### 6.3. Recommendations for Research

Our analysis shows that there is a need to explore the broader potential arising from the use of smartphone apps, including evaluation of the value that can be added to patient pathways. Furthermore, we recommend development of processes to better measure outcomes of pain self-management applications. Feasibility, accessibility and serviceability remain barriers in engaging with such mHealth technologies.

We recommend exploratory research to consider the suitability of different types of smartphone apps or mHealth initiatives with groups of older people to establish reliability, degree of engagement and utility to support the self-management of chronic pain. Understanding how people of all ages and disabilities engage with different healthcare apps should be central to future development of such works.

## 7. Conclusions

Older people can engage with new technology, they should not be excluded from new developments and should be represented in future research. Any future chronic pain app research should have service user engagement utilising a co-design approach, including older people, their carers, pain self-management expertise, health technologists and clinicians, to establish rigour, feasibility, efficacy, utility and engagement.

Without careful consideration of function and usability, acceptance in any population is unlikely. This could be further enhanced with the potential to individualise function within the programming and as technology develops this could be feasible.

The potential for future app development can only increase as the population ages further and the need for remote communication increases. Additionally, smartphone applications have the potential to offer a more personalised, efficient, cost-effective and intelligent pain management experience.

## Figures and Tables

**Figure 1 geriatrics-06-00040-f001:**
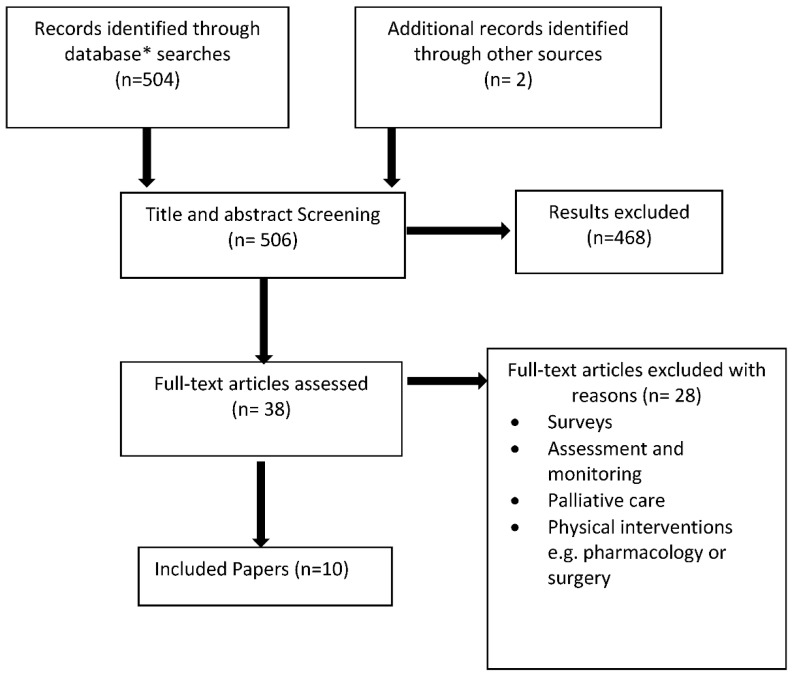
Study selection diagram. * No additional papers were identified from PubMed or Science Direct.

**Table 1 geriatrics-06-00040-t001:** Selection Criteria.

Parameter	Inclusion Criteria	Exclusion Criteria
Population	Studies focused on older adults (≥60 years) or Health Professionals or Carers AND Chronic pain	Younger adults and those without chronic pain, mobility monitoring, surgical intervention, cancer treatment, palliative care or end-of-life support.
Intervention	Telehealth, virtual interventions via phone	Video consultation, instant messaging. Telehealth interventions directed at and solely experienced by health professionals.
Comparator	Any	-
Outcomes	Studies will not be selected on the basis of reported outcomes	-
Study Design	Published primary research studies, including both qualitative and quantitative research.	Non-telehealth related delivery of service research methodologies. Abstracts of unpublished studies. Opinion papers. Professional communications or letters. Literature reviews. Systematic reviews. Meta-analyses. Surveys. Not published in English.

**Table 2 geriatrics-06-00040-t002:** Summary of selected papers.

Author, Country	Year	Study Type	Research Question	Main Findings	Identified App
Bedson, UK [43]	2019	Mixed *n* = 21	To assess face, content and construct validity of data collection using the Pain Recorder in primary care patients receiving new analgesic prescriptions for musculoskeletal pain, as well as to assess its acceptability and clinical utility.	Early testing of an app in a small sample of people consulting with musculoskeletal pain in general practice showed promising results in terms of face and content validity, acceptability and clinical usefulness.	The Keele Pain Recorder- “developed by patients for patients, to improve the management of pain” https://www.keele.ac.uk/kpr/ accessed on 11 March 2021
Bhattarai, Australia [44]	2020	Qualitative, feasibility study *n* = 17	To explore the attitudes and perspectives of primary care and allied health clinicians regarding the integration of pain apps into their older arthritic patients’ pain self- management strategies.	Apps potential to support various aspects of patients’ self-management behaviours.	The DigiTech Pain Project, using the RAISE app
Bhattarai, Australia [45]	2020	Qualitative, feasibility study *n* = 18	To explore the attitudes and experiences of older people with chronic arthritic pain towards using an app for their pain management.	Pain self-management apps have the potential to assist older people in their pain self-management process.	The DigiTech Pain Project, using the RAISE app
Currie, UK [46]	2015	Mixed Method *n* = 168 s	A mixed-methods study of older adults with chronic pain to examine attitudes towards, current use of and acceptance of the use of technology in healthcare.	E-health (including apps) has potential to supplement existing care.	-
Docking, UK [47]	2018	Qualitative (pilot) 24 paramedic students	Usability testing of a newly developed iPhone pain assessment application with potential users.	The pain assessment app constitutes a potentially useful tool (for paramedics) in the prehospital setting for those aged ≥60.	iPhone pain app developed in collaboration with the Computing and Mathematical Sciences (CMS) department at the University of Greenwich.
Levine, USA [48]	2014	Qualitative-(feasibility) focus groups in primary care *n* = 25	To determine how novel telemedicine technologies, particularly smartphones, might be best used in the management of older adults with Chronic Non-Cancer Pain (CNCP).	No participants reported use of telemedicine in geriatric CNCP management. The results suggest that technologies including apps would find a welcome reception among primary care providers delivering care to older adults with CNCP.	-
McDonald (USA) [49]	2013	Pilot for RCT *n* = 23	The more skillful that older adults are in using communication strategies, the more likely they will be to convey important osteoarthritis pain information to practitioners and to be prescribed more effective pain management treatments.	No significant difference in overall pain communication with the practitioner occurred between the pain communication plus virtual pain coach group and the pain communication-only group.	A “virtual pain coach”
Parker (USA)[50]	2013	Qualitative (feasibility) *n* = 41 (older adults)	To examine the willingness of older adults with chronic pain to adopt mHealth technologies.	Older adults with chronic pain are willing and interested in using mHealth including apps.	-
Pimm (UK) [51]	2019	Quasi Experimental *n* = 438	To establish the clinical effectiveness of a web-based pain management programme (PMP), specifically whether it would lead to improved clinical outcomes and reduced healthcare costs in a real-world clinical setting.	A web-based pain management programme can be clinically effective and may be a useful addition to the treatments offered by pain management services. The older participants (50–90 years) were more likely to engage with and complete the programme.	“Pathway through Pain” app https://www.pathwaythroughpain.com/ accessed on 11 March 2021
Richardson (USA) [52]	2018	*n* = 13 Qualitative	To determine the role that smartphones (apps) may play in supporting older adults with chronic noncancer pain (CNCP) in order to improve pain management in this expanding population.	Smartphone apps should support older adult needs to effectively communicate pain experiences with personal contacts and caregivers, as well as healthcare providers.	-

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
