# Peer review of "Smartphone Applications Designed to Improve Older People’s Chronic Pain Management: An Integrated Systematic Review"

_geriatrics, 2021, doi:10.3390/geriatrics6020040_

Round 1
Reviewer 1 Report
The authors conducted a systematic review on a timely topic given the widespread use of smartphone applications for health purposes, among which the self-management of chronic pain is one of the important uses.
The manuscript needs more work to strengthen it. Please see my comments below.
Introduction and Background
The significance and rationale of conducting the systematic review on the selected topic should be addressed more in-depth. For instance, the discussion about the prevalence of smartphone applications and adoption among the target populations. It is nice to mention COVID-19 pandemic to highlight the need of mobile health, but the need of conducting this systematic review is not well justified.
Methods
This section seems to miss some important elements and details. There are no elaborations on the inclusion and exclusion criteria, so the reviewer was not very clear how the articles were selected. Also, how the methodological quality was assessed and scored using nine criteria was not addressed.
The inconsistency of the target populations was spotted. In Table 1, there is “over 65”, but in the later discussion of the reviewed articles, the age groups have various ranges. The authors want to synthesize their review criteria.
The authors included previous review articles in their systematic review which are not appropriate. A systematic review should be on primary literature.
Discussion and Conclusions
The writing in the discussion section reads more like bullet points which should be further consolidated. The barriers of using smartphone applications were not well addressed.
Reviewer 2 Report
Thank you for the opportunity to review this systematic review of papers evaluating smartphone applications designed to improve older people’s chronic pain management. The topic important and relevant.
A systematic review sets out to answer a question. What was the question? The aim was “to evaluate digital health technology initiatives for the management of chronic pain and identify interventions designed to improve management of chronic pain and identify interventions designed to improve older people’s pain management across care settings.” The objectives are different “To asses the impact of telehealth in the form of smartphones, on self-care, to support chronic pain management. To identify the elements of self-care support for chronic pain that can be delivered by telehealth via smartphone intervention.”
This then restricts the review to interventions that are based on smartphones. As the title refers to smartphone applications, the initial implication is that these intervention would be smartphone applications or “apps”, and technically any software running on a mobile phone is an app. Smartphones can and have been used in many ways in telemedicine, from voice calls, video calls, instant messaging (text, audio, video), accessing the Internet and Web sites, and running specific standalone health applications.
What is needed is a more explicit explanation of what is being reviewed, what sort of smartphone use is being considered and for what purpose. There is a need to differentiate between standalone pain management apps designed for self-use, with the information stored on the phone. Those in which data are acquired and transmitted to a service that stores information for serial monitoring and possible automated response or electronic response from a health worker, and those in which there is an expectation of personal response in the event of a worsening situation. All of these options are and have been used for telemedicine. What problem was this systematic review actually aiming to answer?
It would seem that the content and scope of the papers was too disparate to allow for ready classification or grouping.
The inclusion criteria are very broad, English-language studies with human adult participants. No exclusion criteria are stated in the paper other than some of the reasons given for excluding papers in figure 1. They should be given.
Methods, line 79. Results are provided from three of the four databases searched. Presumably, this was because all the papers in Science Direct were duplicates. What was the rationale for choosing to search Medline rather than PubMed which has 20% more references and what was the reason for combining the results from Medline and CINHAL?
The last sentence of the results stating that no papers were found that dealt exclusively with the use of smartphone app technology for pain management in older people should come immediately after Figure 1. If this is not done, the reader is left with a sense of confusion when reading Table 2, as 7 of the 16 papers are not about older people.
The fact that there is literature dealing specifically with the elderly is in effect the key finding of the study. This then highlights the need for further work and opens the door to suggesting what sort of smartphone applications would be of most use.
It needs to be pointed out in the discussion that the findings of this review pertain to developed world. The statement made in line 270, that older people are in the main technologically literate is not borne out by most current literature reporting developing world experience.
Round 2
Reviewer 2 Report
Thank you for opportunity to review this revised paper. As the authors have altered their search strategy, found fewer papers and did not respond in detail to the previous queries I will treat this as a new review. I note as before that this is an important and relevant topic.
There are some fundamental issues with the paper as it currently stands. The title refers to “A systematic review, to evaluate smartphone applications designed to improve older people’s chronic pain management.” The stated aim, is “to consider the existing evidence pertaining to the benefits of smartphone applications for the management of chronic pain in older people”.
We need to look at the definitions of systematic reviews and evidence.
In a systematic review, the approach to answering the research question involves a priori identification of the data or information to be extracted from the identified papers. If this was done it has not been reported. The paper provides a narrative review of papers found in the literature, using an adequately described systematic search method. The narrative review leads to the identification of four themes which then form the focus of the presented results and discussion. The paper does not meet the standard descriptions or definitions of a systematic review. In effect, it is probably a scoping review, see Munn Z et al, BMC Medical Research Methods 2018 and Arksey and O’Malley, 2005.
Of the four themes identified only one is related to the aim of the study – the existing evidence pertaining to the benefits of smartphone applications for the management of chronic pain in older people. What does evidence mean? Evidence is defined in several online dictionaries as: the available body of facts or information indicating whether a belief is true or valid. No quantitative data are available or presented for meta-analysis and the bulk of the information presented is based on the perceptions of health professionals and some older people on the potential benefits of using smartphone apps for pain care and management in older people. As such, the research question cannot be answered based on the data presented, unless the concept of perception of potential benefit equalling fact, is adopted.
Of the 11 papers reviewed, only 5 appear to refer to an app, and fewer to actual smartphone app use by older people.
As a starting point, the authors should reconsider the title of the paper, an option being, “A scoping review, to evaluate the potential use of smartphone applications designed to improve older people’s chronic pain management.”
Other queries and comments.
Background, line 113: the title of the paper is about smartphone application use, but this sentence includes “or similar telehealth initiatives”. If this is correct, the search terms are not adequate and the title is wrong.
The search terms are very different to the previous version of the paper and no longer include the terms smartphone or app. Were terminology alternatives, as provided in Table 1 of the previous paper used? If so, this should be stated.
Section 4.2, untitled table: in the row “Population” and in row 164:, the inclusion criterion is “mean age 60”. Presumably, this should be 60 or more.
Section 4.2, untitled table: What was the rationale for including published research protocols that provide no results and thus no evidence?
Section 4.2, untitled table: literature reviews and systematic reviews are listed as exclusion criteria, yet in line 172, relevant literature reviews were included in hand searching.
What was the actual number of papers found? PRISMA figures usualy report the total number of papers found and then the number of papers after duplicates were removed.
The process of reducing 506 papers to 39 is not described. Who did this and how was it done?
Figure 1: what was the rationale for excluding surveys and assessment and monitoring? If they were about the use of smartphone apps for pain management in the elderly they would be appropriate. Also, these are not listed as exclusion criteria in the table in section 4.2.
Line 848: while the authors of the cited paper may consider an advantage of smartphone telehealth interventions to be anonymity the authors should reflect on this and point out that if apps are used to direct or amend treatment, then the healthcare professional is legally obligated to maintain a record. How are they to do so if they don’t know who the patient is?
In effect, there is very little hard evidence of the worth of smartphone apps for the management of chronic pain in older people.
The topic is important and relevant. The title of the paper and its content are not congruent. Correcting this, should not be difficult.
